# ADAPTIVE SAMPLING SCHEDULER

## ABSTRACT

Consistent distillation methods have evolved into effective techniques that significantly accelerate the sampling process of diffusion models. Although existing methods have achieved remarkable results, the selection of target timesteps during distillation mainly relies on deterministic or stochastic strategies, which often require sampling schedulers to be designed specifically for different distillation processes. Moreover, this pattern severely limits flexibility, thereby restricting the full sampling potential of diffusion models in practical applications. To overcome these limitations, this paper proposes an adaptive sampling scheduler that is applicable to various consistency distillation frameworks. The scheduler introduces three innovative strategies: (i) dynamic target timestep selection, which adapts to different consistency distillation frameworks by selecting timesteps based on their computed importance; (ii) Optimized alternating sampling along the solution trajectory by guiding forward denoising and backward noise addition based on the proposed time step importance, enabling more effective exploration of the solution space to enhance generation performance; and (iii) Utilization of smoothing clipping and color balancing techniques to achieve stable and high-quality generation results at high guidance scales, thereby expanding the applicability of consistency distillation models in complex generation scenarios. We validated the effectiveness and flexibility of the adaptive sampling scheduler across various consistency distillation methods through comprehensive experimental evaluations. Experimental results consistently demonstrated significant improvements in generative performance, highlighting the strong adaptability achieved by our method.

## 1 INTRODUCTION

Diffusion models (Sohl-Dickstein et al., 2015; Song & Ermon, 2019; Ho et al., 2020; Song et al., 2020; Karras et al., 2022; Rombach et al., 2022) have achieved state-of-the-art performance in image generation by effectively modeling complex data distributions and supporting sophisticated conditional mechanisms, such as free-form text prompts. Compared to generative adversarial networks (GANs) (Goodfellow et al., 2014; Karras et al., 2019) and variational autoencoders (VAEs) (Kingma et al., 2013; Sohn et al., 2015), diffusion models employ an iterative denoising procedure that incrementally transforms Gaussian noise into realistic images. Nevertheless, this iterative process typically involves hundreds or thousands of denoising steps, leading to significant computational costs that hinder practical applications.

To overcome these computational limitations, several methods have been proposed to enhance sampling efficiency. These approaches include: (i) accelerating the denoising process by improving ODE solvers (Ho et al., 2020; Lu et al., 2022; 2025); (ii) leveraging knowledge distillation techniques (Salimans & Ho, 2022; Meng et al., 2023) to condense pretrained diffusion models into fewer-step or even single-step generation networks. Recently, consistency models were introduced by Song et al. (2023) as a promising strategy to accelerate image generation. Subsequently, an increasing number of studies have explored consistency distillation methods (Song et al., 2023; Luo et al., 2023; Kim et al., 2023; Wang et al., 2024; Zheng et al., 2024; Wang et al., 2025), which have proven effective in accelerating generation without compromising image quality. These methods utilize a self-consistency property that regularizes predictions of adjacent timesteps to converge toward the same target timestep. Consistency distillation methods are generally classified into two categories based on the strategy used to select the target timestep: (i) Deterministic-target distillation and (ii) Stochastic-target distillation, as illustrated in Figure 1a and Figure 1b.

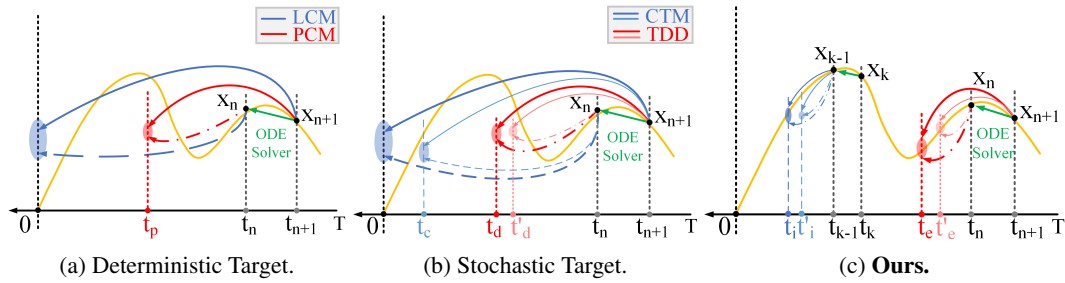

Figure 1: We define existing methods into two categories: (a) Deterministic Target; (b) Stochastic Target. And, the (c) is our **Adaptive Sampling Scheduler** (Deterministic-Stochastic Target).

Deterministic-target distillation employs a fixed mapping pattern to consistently select the same target timestep throughout training, mapping each timestep on the PF-ODE trajectory (Song et al., 2021) to a predetermined target timestep. Early approaches (Song et al., 2023; Luo et al., 2023) predominantly chose the final timestep (0) as the target, resulting in substantial accumulated errors due to long-distance skip predictions. To mitigate this issue, Wang et al. (2024) partition the trajectory into shorter sub-trajectories, using each sub-trajectory's endpoint as the target timestep to reduce the error caused by extensive skip predictions. However, the fixed sub-trajectory lengths limit adaptability to varying inference step counts.

Stochastic-target distillation, conversely, utilizes a one-to-many random mapping strategy, assigning each current timestep to a randomly selected future timestep (Kim et al., 2023; Zheng et al., 2024). This method allows training to generalize across different schedules effectively. Nevertheless, it usually demands significant computational resources. Recently, Wang et al. (2025) aims to reduce training overhead by randomly selecting target timesteps from a predefined set, effectively balancing performance and computational efficiency, but the need to set the predefined set in advance limits its generality.

Although these methods have demonstrated promising results, we observed notable limitations stemming from their individualized strategies for selecting target timestep patterns. Specifically, most existing approaches rely on customized sampling schedulers, and their performance substantially deteriorates when applied to general sampling schedulers. Moreover, severe exposure issues arise at higher guidance scale values.

To overcome these issues, we analyzed the diffusion process and identified that the rate of change in the Signal-to-Noise Ratio (SNR) varies distinctly at each timestep along the trajectory. Motivated by this observation, we propose a novel universal **Adaptive Sampling Scheduler** that leverages the timestep-specific SNR change rate as a criterion for determining the target timestep. This scheduler effectively generalizes across various consistency distillation methods, yielding improved sampling outcomes. Additionally, to mitigate exposure issues at higher guidance scale values, we introduce smoother clipping and color balancing techniques, further enhancing the generation quality.

MAIN CONTRIBUTIONS

- We propose a more reasonable criterion (**Importance**) for selecting the target timestep **based on the rate of change in the signal-to-noise ratio (SNR)**, combining deterministic-target and stochastic-target.

- We propose **Adaptive Sampling Scheduler**, which introduces a new target timestep sampling scheduling strategy (Deterministic-Stochastic Target) **based on the importance of timesteps**. At the same time, we better mitigate the exposure problem of high guidance scale values through smoothing processing of the sampling process clipping method and color balance method.

- Experiments show that ours provides a more general and reasonable sampling scheme for consistency distillation methods, further improving the performance of the generation task.

## 2 RELATED WORK

Diffusion models achieve state-of-the-art image generation by iteratively denoising noisy inputs (Sohl-Dickstein et al., 2015; Song & Ermon, 2019; Ho et al., 2020; Song et al., 2020; Rombach et al., 2022), surpassing VAEs and GANs (Kingma et al., 2013; Sohn et al., 2015; Goodfellow et al., 2014; Karras et al., 2019). However, their multi-step refinement incurs substantial computational cost, hindering deployment in latency-sensitive or real-time applications. This trade-off between fidelity and speed has spurred the search for more efficient sampling paradigms.

In response, Consistency Models (CM) (Song et al., 2023) have emerged as a promising solution. By learning a mapping that projects any point along the diffusion ODE trajectory back to the original data manifold, CMs enable few- or even single-step sampling without degrading image quality. Moreover, they can be trained via knowledge distillation from powerful pretrained diffusion networks or learned independently, offering flexibility across different use cases. Building on this foundation, numerous consistency distillation methods have been proposed to further optimize efficiency and performance. Luo et al. (2023) employ skip predictions to accelerate generation within latents, while PCM (Wang et al., 2024) partitions the ODE path into sub-trajectories and uses each endpoint as the distillation target. Fixed-target schemes, however, lack adaptability to varying samplers; approaches like CTM (Kim et al., 2023) and TCD (Zheng et al., 2024) introduce random jumps but compromise training efficiency. To strike a better balance, TDD (Wang et al., 2025) selects sub-target timesteps randomly from a predefined set, achieving less training cost.

## 3 PRELIMINARIES

### 3.1 DIFFUSION MODEL

Diffusion models (Sohl-Dickstein et al., 2015; Song & Ermon, 2019; Ho et al., 2020; Song et al., 2020), or score-based generative models (Song et al., 2021), represent a family of generative models that draw inspiration from the principles of thermodynamics and stochastic processes. These models involve the gradual injection of Gaussian noise into data, followed by the generation of samples from the noise through a process of reverse denoising. Let $p_{data}(x)$ denotes the origin data distribution and $p_t(x)$ is the distribution of $x$ at time $t$, where $\{x_t | t \in [0, T]\}$. From a continuous-time perspective, the forward process can be described by a stochastic differential equation (SDE) (Song et al., 2021; Lu et al., 2022; Karras et al., 2022). The stochastic trajectory is described by the following equation:

$$\mathrm{d}x_t = f(t)x_t\,\mathrm{d}t + g(t)\,\mathrm{d}w_t, \quad x_0 \sim p_{data}(x_0) \tag{1}$$

$$f(t) = \frac{\mathrm{d}\log\alpha_t}{\mathrm{d}t}, \quad g^2(t) = \frac{\mathrm{d}\sigma_t^2}{\mathrm{d}t} - 2\frac{\mathrm{d}\log\alpha_t}{\mathrm{d}t}\sigma_t^2 \tag{2}$$

where $w_t$ is the standard Brownian motion, and $\alpha_t, \sigma_t$ specify the noise schedule. And $f(t)x_t$ denotes the drift coefficient for deterministic changes, and $g(t)$ is the diffusion coefficient for stochastic variations.

The Probabilistic Flow Ordinary Differential Equation (PF-ODE) (Song et al., 2021; Lu et al., 2022) proposes that diffusion processes described by stochastic differential equations (SDE) can be described in deterministic form using deterministic processes with the same marginal distribution. The PF-ODE is formulated as:

$$\mathrm{d}x_t = \left[ f(t)x_t - \frac{1}{2}g(t)^2 \nabla_x \log p_t(x) \right] \mathrm{d}t \tag{3}$$

where $\nabla_x \log p_t(x)$ is called the *score function*, indicates the gradient of the log density of $p_t(x)$. Empirically, in the standard diffusion training process, we aim to train a score model $s_\phi(x, t)$ to approximate this score function using by score matching, which is equivalent to $s_\phi(x, t) \approx \nabla_x \log P_t(x) = \mathbb{E}_{x_0 \sim P(x_0|x)} [\nabla_x \log P_t(x|x_0)]$, substitue the $\nabla_x \log P_t(x)$ with $s_\phi(x, t)$, and we get the empirical PF-ODE. Despite the plethora of methods such as (Song et al., 2020; Lu et al., 2022; Karras et al., 2022) can approximate ODE solutions, using only a handful of sampling steps (e.g., 4 or 8) inevitably incurs significant discretization errors, leading to unsatisfactory outcomes.

## 3.2 CONSISTENCY MODELS

Consistency Models (Song et al., 2023) constitutes a novel family of generative models capable of one-step or few-step generation by learning a mapping that projects any intermediate points along the PF-ODE trajectory back to the initial point. A consistency model $f_\theta(\cdot, t)$ learns to achieve $f_\theta(x_t, t) = x_\epsilon$ must adhere to the *self-consistency property*:

$$f_\theta(x_t, t) = f_\theta(x_{t'}, t'), \quad \forall t, t' \in [\epsilon, T] \tag{4}$$

where $\epsilon$ is a fixed small positive number. Consistency Models can be trained using pre-trained model distillation or trained from scratch, with the former referred to as consistency distillation.

## 3.3 CONSISTENCY DISTILLATION

For uniformity in subsequent notation, we define $\phi$ to denote the teacher model, $f_\theta$ to denote the student consistency model, and $\Phi$ to denote the selected numerical ODE Solver, and the $\hat{x}_{t_n}^\phi$ is one-step estimation of $x_{t_n}$ from $x_{t_{n+1}}$ by $\Phi$ as follows:

$$\hat{x}_{t_n}^\phi \leftarrow x_{t_{n+1}} + (t_n - t_{n+1})\Phi(x_{t_{n+1}}, t_{n+1}; \phi) \tag{5}$$

To enforce the *self-consistency property*, define the consistency loss as follows:

$$\mathcal{L}_{cm} = \mathbb{E}_{x,t}\left[d\big(f_\theta(x_{t_{n+1}}, t_{n+1}, \tau), f_{\theta^-}(\hat{x}_{t_n}^\phi, t_n, \tau)\big)\right] \tag{6}$$

where $d(., .)$ is a chosen metric function to calculate the distance between two samples, e.g., the squared $\ell_2$ distance. The $f_{\theta^-}$ is the consistency model with a target model updated with exponential moving average (EMA) of the parameter $f_\theta$ we intend to learn, here $\theta^- \leftarrow \mu\theta^- + (1-\mu)\theta$, $\mu = 0.95$, and the $\tau$ refers to the target timestep.

For existing work, deterministic-target distillation method CM (Song et al., 2023) set $\tau = 0$ for any timestep $t_{n+1}$, and LCM (Luo et al., 2023) set $\tau = t_n$ to achieve the skip prediction, drastically reducing the length of time schedule from thousands to dozens. Next, PCM (Wang et al., 2024) divide the entire trajectory into multiple phased sub-trajectories (e.g. 4, 8), select the next phased ending point to be $\tau$. For stochastic-target distillation methods, CTM (Kim et al., 2023) selects a random $\tau$ within the interval $[0, t_n]$, and TDD (Wang et al., 2025) selects a random $\tau \in [(1-\boldsymbol{\eta})t_m, t_m]$ where $t_m \in [t - e, t]$, $t$ is a predefined subset timesteps, the $e, \eta$ are preset hyper-parameters.

We found that previous studies either used a fixed target timestep or a random target timestep. They lacked an criterion for selecting the target timestep. Therefore, we considered *How to more reasonably select the target timestep in a standardized manner*?

## 4 IMPORTANCE OF TIMESTEPS

For the sake of this concern, we first review the forward diffusion process illustrated in Figure 2. The visualization makes it clear that, up to $T = 200$, the image content remains almost fully

T=0  T=100  T=200  T=400  T=500  T=600  T=700  T=800  T=900  T=999

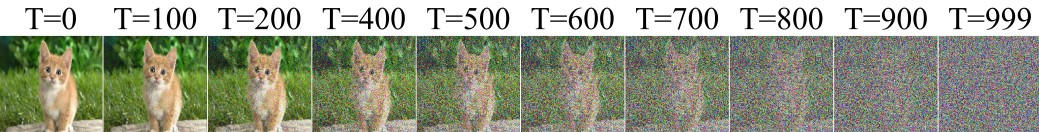

Figure 2: Forward diffusion results at some timesteps by DDPM (Ho et al., 2020).

discernible, while beyond $T = 800$ it becomes virtually unrecognizable. In these two regions, the signal retention rates are respectively very high and very low, and the visual changes from step to step are minimal. In contrast, during the intermediate phase ($T = 400 \rightarrow 700$), the images undergo the most significant transformations, reflecting a rapid degradation of detail. Therefore, we further defined the rate of signal change using Equation 7. We call it "Importance ($I$)". We calculated the importance of all timesteps, and the visualization results are shown in Figure 3a. Additionally,

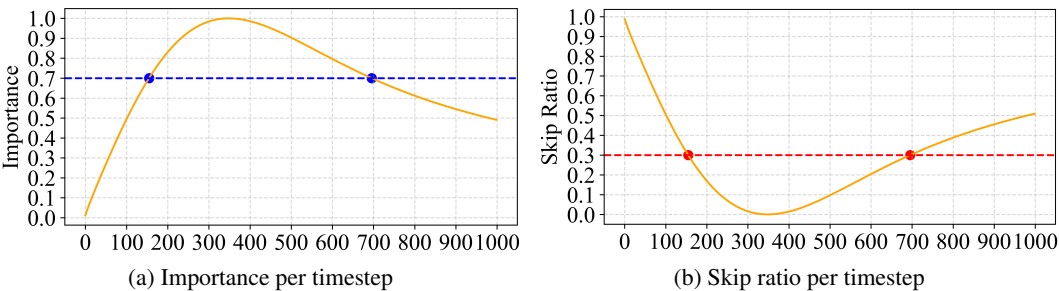

(a) Importance per timestep

(b) Skip ratio per timestep

Figure 3: Importance and skip ratio across timesteps in the diffusion process.

we defined the skip rate $R$ to assist in controlling the forward and backward jumps mentioned in Equation 10, as shown in Figure 3b.

$$I_t = \frac{\left|\nabla_t \ln\left(\frac{\overline{\alpha}_t}{1 - \overline{\alpha}_t} + \varepsilon\right)\right|^{-1}}{\max\limits_{0 \leq j < T} \left|\nabla_j \ln\left(\frac{\overline{\alpha}_j}{1 - \overline{\alpha}_j} + \varepsilon\right)\right|^{-1}}, \overline{\alpha}_t = \prod_{i=1}^{t} \alpha_i; \quad R_t = 1 - I_t + \varepsilon \tag{7}$$

The $\varepsilon = 1e^{-8}$ to avoid division by zero. As can be seen in Figure 3a, the changes in the diffusion process shown in Figure 2 can be reasonably approximated by Equation 7 (*I more closer to 1, the faster the change; more closer to 0, the slower the change*). Through analysis of the diffusion process, we argue that when selecting the target timestep, *not all timesteps should be treated equally*, but rather should depend on the importance of the current timestep. Therefore, based on this finding, we proposed Adaptive Sampling. In previous studies, most work (Song et al., 2023; Luo et al., 2023; Wang et al., 2024) have used equidistant sampling. Wang et al. (2025) mention that extending the sampling method to non-equidistant sampling will yield better sampling results, but it uses predefined timesteps for sampling. In order to address these limitations, thus, we propose adaptive sampling.

## 5 ADAPTIVE SAMPLING

According to Equation 7, we calculate the importance corresponding to all timesteps. We take the timestep with the maximum importance in different intervals as the importance sampling timestep $T_I$. At the same time, we define the original equidistant sampling timestep $T_E$. The equation is defined as follows:

$$T_{as} = \{t_i \mid t_i \in T_I, I_t > \theta\} \cup \{t_i \mid t_i \in T_E, I_t \leq \theta\} \tag{8}$$

and according to Figure 2 and Figure 3a, we set $\theta = 0.7$ as the threshold. For a more intuitive understanding, we illustrate the process in Figure 4a. Here, we finally obtain a set of target timesteps for $T_{as}$, in which the number of target timesteps is still the same as $T_n$, but the intervals between adjacent target timesteps are not the original equal intervals, but have changed. In addition, we referred to the $\gamma$ sampler proposed by Kim et al. (2023), which solves $x_0$ by alternately performing forward and backward jumps on the solution trajectory. The $\gamma$ parameter can be adjusted to control the proportion of randomness (default $\gamma = 0.2$ that is same with CTM (Kim et al., 2023)), which has been proven to improve the generation quality to a certain extent. On this basis, we optimized using importance and replaced the forward and backward jumps based on Equation 8, $\gamma$-I Sampler, as shown in Figure 4b. Here, our method can be easily understood from the Figure 4b. The $\gamma$-I sampler first denoise the current noise sample using the network in each backward, and then reintroduces noise proportionally. The denoising and noise addition process is as follows:

$$t_{n+1} \xrightarrow{\text{Denoise}} \sqrt{(1-\gamma)^2} * t_n \xrightarrow{\text{Noisify}} t_n, \ t_n \in T_E \tag{9}$$

$$t_n \xrightarrow{\text{Denoise}} R_t * t_{n-1} \xrightarrow{\text{Noisify}} t_{n-1}, \ t_{n-1} \in T_I \tag{10}$$

In addition, in all previous methods, when high classifier-free guidance (CFG) (Ho & Salimans, 2022) scaling was used, there were varying degrees of exposure issues. To alleviate this issue,

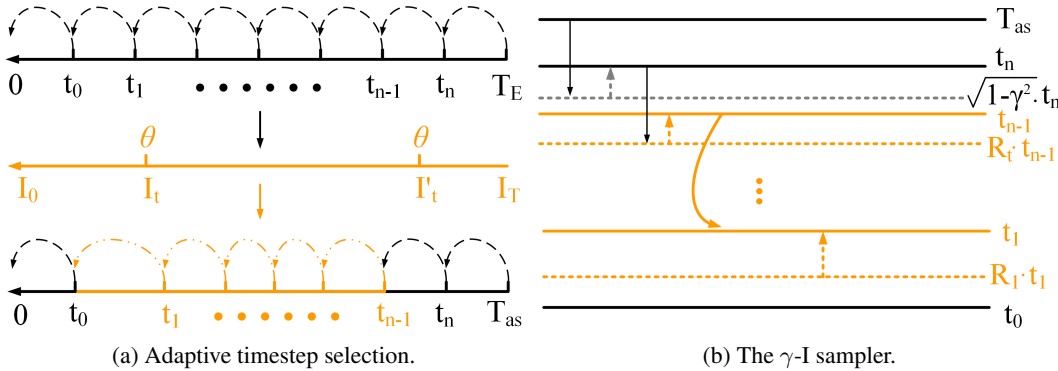

(a) Adaptive timestep selection.  (b) The $\gamma$-I sampler.

Figure 4: Our adaptive sampling process: (a) select timesteps adaptively; (b) apply the $\gamma$-I sampler.

we consulted the solutions offered by Saharia et al. (2022); Lu et al. (2022). We then integrated these solutions into our sampling scheduler after our optimization, which effectively alleviated the exposure issue. Additionally, we implemented a color balance method to assist in generating higher guidance scales. The formula is as follows:

$$x_0 = \frac{e^{x_0} - e^{-x_0}}{e^{x_0} + e^{-x_0}}; \quad x_c = x_c - \alpha \cdot \mathrm{mean}(x_c); \quad x_0 = x_0 - \beta \cdot \mathrm{mean}(x_0) \tag{11}$$

where $x_c$ is the each channel of $x_0$, the $\alpha, \beta = 0.5$. Without changing the shape of $x_0$, we used the hyperbolic tangent function to map all values to the range $(-1, 1)$, thereby removing outliers. Compared to the mapping methods in Saharia et al. (2022); Lu et al. (2022), ours do not require prior deformation and provides a more direct and smooth mapping. Although hyperbolic tangent function compresses values between $(-1, 1)$, if the $x_0$ deviates too much from 0 (e.g. exposure situation), most values will fall into the saturation zone (output tends to $\pm 1$). Therefore, we further offset the mean of $x_0$ within the channel and across the entire image by a certain proportion, so that more values are concentrated in the linear interval of hyperbolic tangent function, thereby retaining more effective information.

## 6 EXPERIMENTS

### 6.1 BACKBONES

We chose text-to-image generation as the basic task for all experimental evaluations. For an objective and comprehensive comparison, we conducted image generation experiments at 1024 resolution and 512 resolution, selecting two different architectures as the backbone for the comparison experiments: Stable Diffusion XL (SDXL) (Podell et al., 2023) for 1024 resolution and Stable Diffusion v1-5 (SD v1-5) (Rombach et al., 2022) for 512 resolution.

### 6.2 BASELINES & EVALUATION

We choose previous research: LCM (Luo et al., 2023), PCM (Wang et al., 2024), TCD (Zheng et al., 2024) and TDD (Wang et al., 2025) as baselines. All relevant backbone models and baseline models have been open-sourced. The PCM, TCD, TDD are used to generate the resolution of 1024, while LCM, PCM are also used to generate 512 resolution. For performance evaluation, we utilize the validation split of the MS COCO 2014 dataset (Lin et al., 2014), following Karpathy's 30K partition, and to generate image prompts we use the first sentence of each image's default caption. And, for different resolutions of different backbones, we report the key metrics of the generated images, adopt three different metrics to assess our model's outputs: the Fréchet Inception Distance (FID) (Heusel et al., 2017) to measure the distributional similarity between generated and real images, the CLIP Score (Radford et al., 2021) to quantify semantic alignment with input prompts, and the Inception Score (IS) (Salimans et al., 2016) to evaluate both the visual quality and diversity of the generated samples.

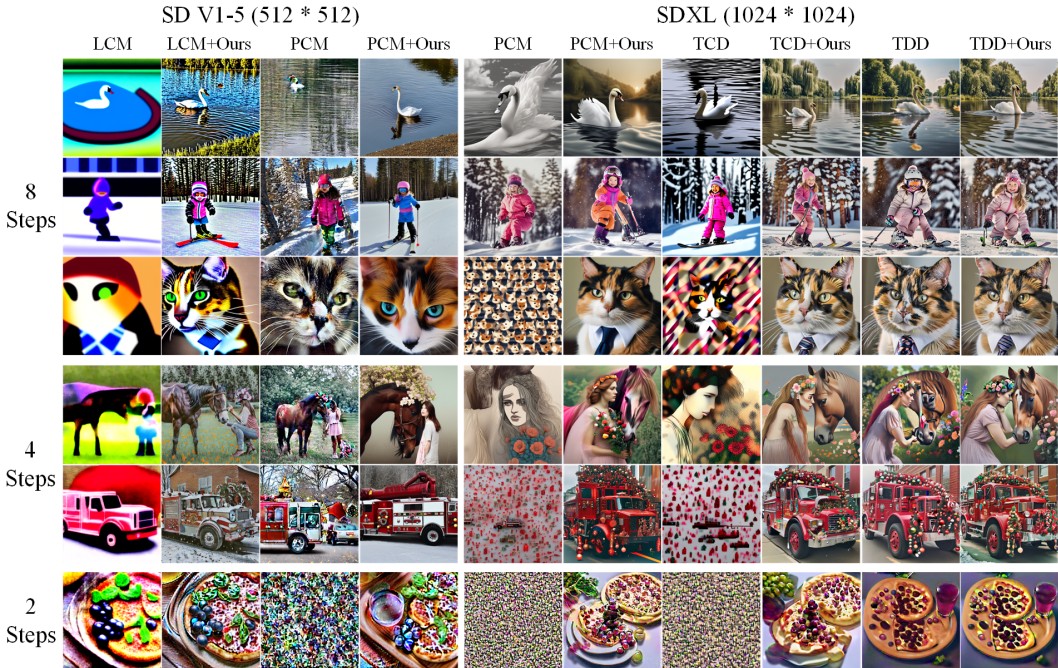

Figure 5: Qualitative comparison of different methods using 2, 4, and 8 steps for two diffusion models: SD V1-5 (Rombach et al., 2022), SDXL (Podell et al., 2023).

Table 1: Performance comparison at 1024 × 1024 resolution using Stable Diffusion XL (Podell et al., 2023), evaluated on FID (lower is better), CLIP Score, and Inception Score (higher is better), with 2, 4, and 8 sampling steps. The Δ denotes Mean Performance Improvement (MPI).

| Methods | FID ↓ | | | CLIP Score ↑ | | | Inception Score ↑ | | |
|---|---|---|---|---|---|---|---|---|---|
| | 2 steps | 4 steps | 8 steps | 2 steps | 4 steps | 8 steps | 2 steps | 4 steps | 8 steps |
| PCM (Wang et al., 2024) | 372.82 | 112.65 | 31.73 | 18.44 | 24.18 | 30.44 | 1.71 | 11.67 | 25.78 |
| PCM + **Ours** | **65.83** | **29.40** | **23.21** | **30.22** | **30.40** | **31.52** | **16.54** | **24.59** | **31.80** |
| TCD (Zheng et al., 2024) | 363.50 | 103.66 | 53.72 | 18.73 | 26.05 | 30.44 | 1.82 | 12.47 | 17.83 |
| TCD + **Ours** | **62.89** | **28.51** | **27.44** | **28.88** | **31.71** | **32.06** | **16.33** | **28.02** | **32.17** |
| TDD (Wang et al., 2025) | 58.45 | 29.80 | 27.72 | 29.27 | 31.12 | 31.47 | 17.18 | 27.02 | 29.72 |
| TDD + **Ours** | **55.71** | **27.88** | **26.60** | **29.49** | **31.47** | **31.74** | **17.62** | **29.36** | **32.88** |
| Δ (MPI) | 203.45 | 53.44 | 11.97 | 7.38 | 4.08 | 0.99 | 9.93 | 10.27 | 7.84 |

## 6.3 MAIN RESULTS

The quantitative results in Table 1 and Table 2 demonstrate that our method consistently outperforms the baseline method across both SDXL and SD v1-5. Notably, there are significant performance gains in the smaller step (e.g. 2 or 4), highlighting the efficiency and superiority of our approach.

As can be seen from Table 2, the LCM (Luo et al., 2023) showed a counterintuitive experimental phenomenon at 4 steps and 8 steps. When the number of steps was larger, the FID and IS evaluations showed a decline. Through experimentation, we found that this is because in the original LCM method, distillation is performed using relatively large CFG values during training, so when large CFG values are used in inference, the more steps there are, the more serious the exposure issue becomes. Ours can still greatly alleviate this issue.

From a qualitative standpoint, Figure 5 vividly illustrates our method's prowess under extreme sampling constraints (2 or 4 steps, CFG = 7.5): whereas the SDXL and SD v1-5 baselines produce nothing more than chaotic noise and meaningless textures, our approach consistently reconstructs coherent, high-fidelity images even with only two steps.

Table 2: Performance comparison at 512 × 512 resolution using Stable Diffusion v1-5.

| Methods | FID ↓ | | | CLIP Score ↑ | | | Inception Score ↑ | | |
|---|---|---|---|---|---|---|---|---|---|
| | 2 steps | 4 steps | 8 steps | 2 steps | 4 steps | 8 steps | 2 steps | 4 steps | 8 steps |
| LCM (Luo et al., 2023) | 86.33 | 88.20 | 109.84 | 28.02 | 26.48 | 25.19 | 14.28 | 11.73 | 8.71 |
| LCM + **Ours** | **58.01** | **30.04** | **47.67** | **30.18** | **30.71** | **30.79** | **17.18** | **28.81** | **19.84** |
| PCM (Wang et al., 2024) | 424.04 | 89.99 | 38.82 | 18.99 | 26.51 | 30.02 | 1.76 | 12.49 | 21.07 |
| PCM + **Ours** | **60.66** | **23.11** | **22.47** | **29.93** | **30.37** | **31.03** | **16.86** | **28.93** | **31.65** |
| Δ (MPI) | 195.85 | 62.52 | 39.26 | 6.55 | 4.05 | 3.31 | 9.00 | 16.76 | 10.86 |

Quantitatively, Table 1 confirms this advantage, with baseline FID scores skyrocketing into the hundreds at 2 or 4 steps, whereas our scheduler brings FID down to practical levels across 2, 4 steps, demonstrating both the robustness and superiority of our method in low-budget sampling scenarios.

## 6.4 ABLATION STUDY

In order to gain a more comprehensive understanding of our approach, we conducted a series of detailed ablation experiments on the methods proposed in our paper, select TDD (Wang et al., 2025) as the baseline.

### 6.4.1 DIFFERENT IMPORTANCE VALUES

We selected different values of $\theta$ and conducted further comparative experiments. As shown in

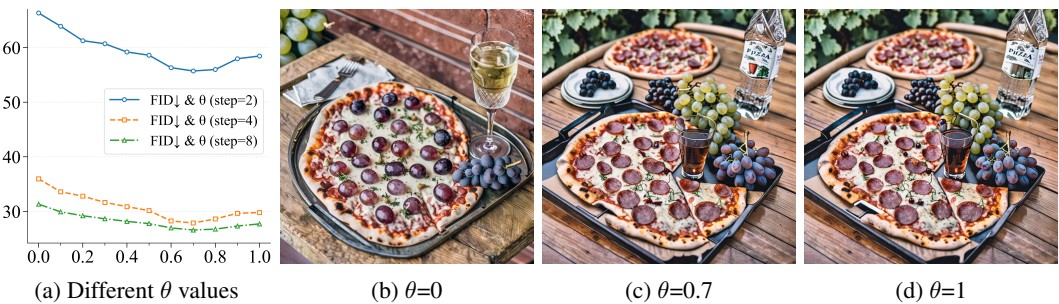

| (a) Different $\theta$ values | (b) $\theta$=0 | (c) $\theta$=0.7 | (d) $\theta$=1 |
|---|---|---|---|

Figure 6: Results of different $\theta$ values. Prompt: *A pizza and grapes sit on a tray next to a drink.*

Figure 6, we can see that when $\theta = 0$, timesteps are chosen purely by importance, yielding images that are more random yet still retain overall structure. In contrast, when $\theta = 1$, sampling proceeds at fixed intervals, this produces outputs that more faithfully follow the prompt but introduces structural ambiguity *(e.g., the wine glass and grapes appear to merge)*. When $\theta = 0.7$, the image structure is clear and consistent with the prompt, and the font on the bottle is clearer and the colors are richer. The results of ablation in Figure 6a is consistent with the importance curve shown in Figure 3a, further proving that the *Importance* we propose is reasonable and effective.

### 6.4.2 THE $\gamma$-I SAMPLER

We compare it with the original $\gamma$ sampler proposed by CTM (Kim et al., 2023) and one that does not use the $\gamma$ sampler. The results in Figure 7 show that without using the $\gamma$ sampler, the generated structure is the worst *(e.g., with three paws and two tails)*. Using the $\gamma$ sampler improves the situation, and when using our proposed $\gamma$-I sampler, the structure is the most reasonable, the actions are more consistent with the prompt (*eating fruit*) and more details in fruits.

### 6.4.3 SMOOTH CLIPPING AND COLOR BALANCE

In order to verify the effectiveness of the proposed smoothing clipping and color balancing techniques, all parameters of the other methods proposed in this paper were fixed and compared with the previous clipping methods. All results are obtained by using CFG = 7.5 and 8 steps.

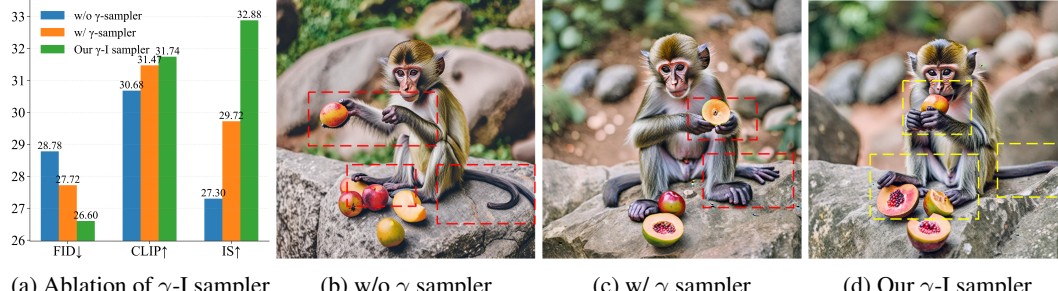

(a) Ablation of $\gamma$-I sampler     (b) w/o $\gamma$ sampler     (c) w/ $\gamma$ sampler     (d) Our $\gamma$-I sampler

Figure 7: Prompt: *Small monkey eating fruit sitting on a rock*. Using CFG = 7.5 and 8 steps.

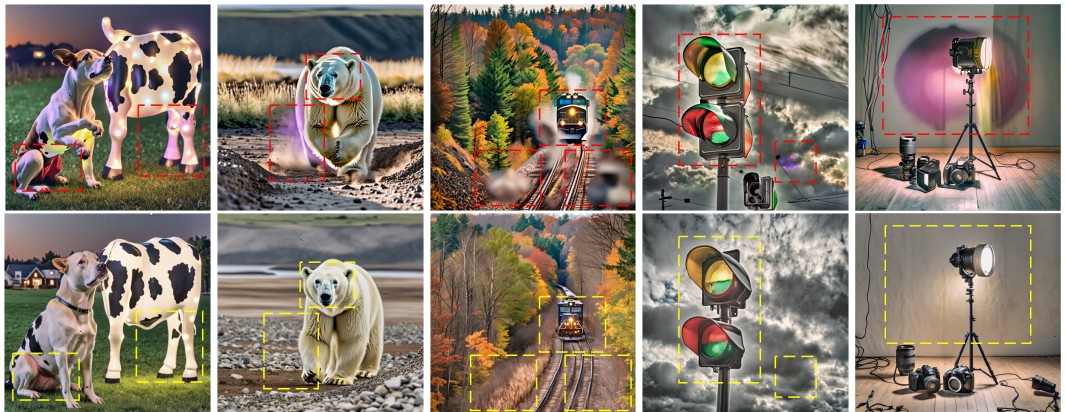

Figure 8: The first row **w/o** ours, using the clipping method same with Saharia et al. (2022); Lu et al. (2022), and the second row **w/** ours. We circled some obvious areas in the picture.

The experimental results are presented in Figure 8. Our approach markedly alleviates color overexposure, yielding cleaner and more natural images. Previous consistency distillation methods often suffered from pronounced overexposure caused by classifier-free guidance (CFG) scaling during distillation. Consequently, these methods typically resorted to low guidance scales (e.g., CFG = 1 or 2) at sampling. By incorporating the strategies of Saharia et al. (2022); Lu et al. (2022), we refined their clipping procedures and introduced a dedicated color-balancing step, which significantly suppresses overexposure artifacts and improves overall color fidelity. More detailed discussion of the issue of exposure to high CFG values is provided in the Appendix A.

## 7 CONCLUSION

We introduce a novel, universally applicable adaptive sampling scheduler grounded in consistency distillation, designed to overcome the key limitations of previous deterministic or stochastic target strategies. By dynamically selecting target timesteps based on their computed importance, quantified via the rate of change in signal-to-noise ratio (SNR), our scheduler adaptively focuses computation on the most critical diffusion steps, meanwhile, we further optimize the alternating forward and backward jumps according to timestep importance, substantially enhancing generation quality across diverse consistency distillation methods. And, employ a combination of smoothing clipping and color balancing to further mitigate exposure artifacts at high guidance scales. Extensive experiments on standard SDXL and SD v1-5 benchmarks at multiple resolutions confirm the effectiveness and robustness of our method. Moreover, our scheduler can seamlessly integrates with existing consistency distillation frameworks, further underscoring its practicality. We hope these insights will propel further advances in fast, high-quality generative sampling.

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
