# A APPENDIX

## A.1 EXPOSURE PROBLEM ANALYSIS

In text-conditioned diffusion, a neural network $\epsilon_\phi$ is trained to predict the noise at each timestep $t$ given a noisy $x_t$ and a positive prompt embedding $c$. By also training with $c$ replaced by the null embedding $\varnothing$, the model acquires an unconditional predictor $\epsilon_\phi(x_t, t, \varnothing)$. At inference, Classifier-Free Guidance (Ho & Salimans, 2022) interpolates these two estimates using a guidance scale $\omega \geq 0$:

$$\tilde{\epsilon}_\phi(x_t, t, c; \omega) = (1 + \omega)\, \epsilon_\phi(x_t, t, c) - \omega\, \epsilon_\phi(x_t, t, \varnothing) \tag{12}$$

where $\omega = 0$ yields pure conditional sampling and larger $\omega$ amplifies the conditional signal. This simple mechanism requires no auxiliary classifier, affords a smooth fidelity-diversity trade-off via $\omega$, and has been instrumental in models such as Stable Diffusion (Rombach et al., 2022), Imagen (Saharia et al., 2022).

Define $\epsilon_\theta(x_t, t, c)$ as a consistency model is trained to match the ODE trajectory in a single or few steps. Naively solving the diffusion ODE with pure conditional predictions $\epsilon_\theta(x_t, t, c)$ leads to trajectories that deviate from the data manifold, yielding blurred or structurally inconsistent outputs. Hence, each step employs a CFG-augmented estimate that may also incorporate a null or negative embedding $c_n$:

$$\epsilon_\theta(x_t, t, c, c_n; \omega) = \epsilon_\theta(x_t, t, c_n) + \omega\big(\epsilon_\theta(x_t, t, c) - \epsilon_\theta(x_t, t, c_n)\big) \tag{13}$$

At inference, if we further apply CFG with scale $\omega'$ to the consistency model itself. Denoting by $x_t^{\omega'}$ the state distilled under diversity scale $\omega'$, its noise prediction becomes:

$$\hat{\epsilon} = (1 + \omega)\, \epsilon_\theta\big(x_t^{\omega'}, t, c\big) - \omega\, \epsilon_\theta\big(x_t^{\omega'}, t, \varnothing\big) \tag{14}$$

As theoretical analysis in PCM (Wang et al., 2024) shows that this compounds the consistency model: for all $t' \leq t$, we have:

$$\epsilon_\theta(x_t, t, c, c_n; \omega') \propto \omega\, \omega'\big(\epsilon_\phi(x_{t'}, t', c) - \epsilon_m^\phi\big) + \epsilon_\phi(x_{t'}, t', c_n) \tag{15}$$

where $\epsilon_m^\phi = (1 - \alpha)\, \epsilon_\phi(x_{t'}, t', c_n) + \alpha\, \epsilon_\phi(x_{t'}, t', \varnothing)$ and $\alpha = (\omega - 1)/(\omega\, \omega')$. This is equivalent to scaling the predictions of the original diffusion model by a factor of $\omega\, \omega'$, which leads to severe exposure issues when using larger CFG values during inference. Thus, large CFG (e.g., CFG > 7) accentuates edges and contrast but suppresses texture complexity and flattens fine details, whereas setting CFG (e.g., CFG = 1) too low causes the ODE path to drift off-manifold, producing blurry or semantically inconsistent samples.

In addition, we observe that prior work has typically proposed its own solutions. For example, PCM (Wang et al., 2024) trains and stores model weights for various values of $\omega$ to accommodate subsequent applications with different $\omega'$ for inference; TCD (Zheng et al., 2024) provides weights that perform well when CFG is around 2.5 but, as our experiments show in Table 1 and Figure 5, degrade severely when CFG > 7; and Wang et al. (2025) mitigates the exposure problem by fixing CFG at 3.5 during training, and using the clipping method proposed by (Saharia et al., 2022; Lu et al., 2022), yet the exposure issue still occurs when CFG (e.g. 7.5) is higher or when the number of inference steps is increased (e.g. to 8 steps).

In practice, most diffusion models employ a CFG scale of 7.5 to achieve outputs that adhere more closely to the prompt. Therefore, we did not deliberately choose a smaller CFG value; instead, to remain consistent with real-world usage, all of our experiments were conducted with CFG set to 7.5. Using our method, the severe overexposure issues induced by large CFG values (i.e. CFG > 7) can be effectively mitigated.

## A.2 SMOOTH CLIPPING & COLOR BALANCE

When employing a high CFG scale (e.g. $\omega = 7.5$) together with more inference steps (8 steps), most existing methods exhibit severe over-exposure artifacts. Even applying the cropping strategy of prior work (Saharia et al., 2022; Lu et al., 2022) only partially mitigates the effect-the exposure problem persists.

In an over-exposed image, the vast majority of pixel magnitudes cluster near the maximum, so the quantile-based threshold:

$$s = \text{Clamp}\left(\text{quantile}_\alpha(|x|),\, 1,\, V\right) \qquad (16)$$

(with, e.g., $\alpha = 0.995$) almost always saturates at its upper bound $V$. All Pixels are then normalized via follow:

$$\tilde{x}_i \;=\; \frac{\text{clip}(x_i,\, -s,\, s)}{s} \qquad (17)$$

which, when $s = V$, reduces to $\tilde{x}_i = \frac{x_i}{V}$. Under this mapping, even mid-tone values (e.g. $x_i \approx 0.5V$) are confined to the narrow interval $[0.5, 1]$, collapsing the dynamic range, obliterating contrast, and flattening image details.

By contrast, our proposed $\tanh(x)$ and color balance scheme overcomes these issues in two ways. First, $\tanh(x)$ enforces soft saturation it smoothly compresses all values into $(-1, 1)$ and only gradually attenuates extremes, rather than hard-clipping them. This preserves continuous saturation transitions and avoids abrupt clipping artifacts. Second, if the $x_0$ deviates too much from reasonable area (e.g. exposure situation), most values will fall into the saturation zone (output tends to $\pm 1$) as clamp in (Saharia et al., 2022; Lu et al., 2022), so continue subtracting the per-channel and then the global mean recenter the data around reasonable area, maintaining relative brightness differences across regions without uniformly suppressing low-level and mid-level intensities through a single global threshold.

### A.3 MORE EXPERIMENTAL RESULTS

To further demonstrate the effectiveness and robustness of our method, we have added additional visual comparisons in Figure 9, Figure 10 and Figure 11. Meanwhile, more generation results of ours are shown in Figure 12. Here, we select the original form of the previous methods, and combined them with ours to clearly highlight the improvements of our solution in terms of detail preservation, exposure correction, and overall visual fidelity.

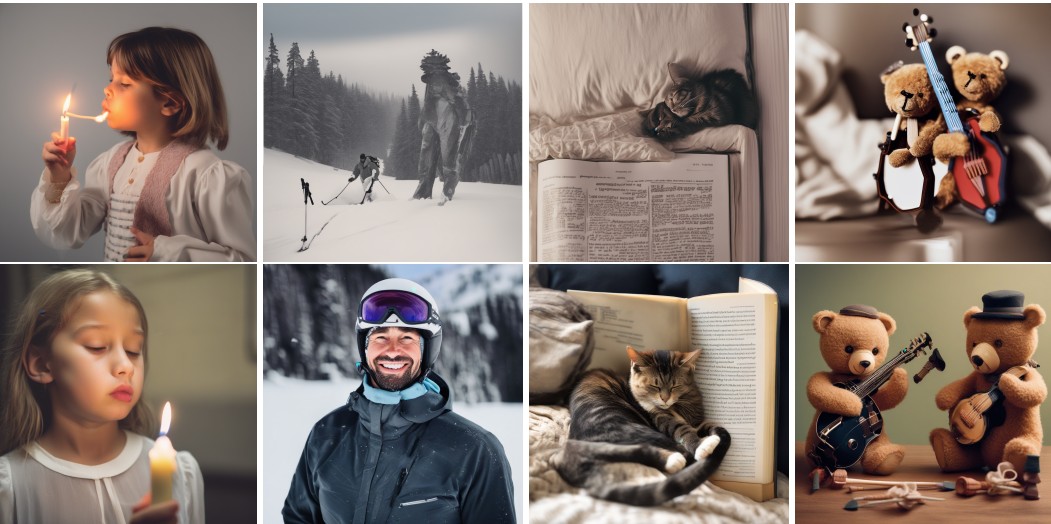

Figure 9: Qualitative comparison of PCM (Wang et al., 2024) based on Stable Diffusion XL. Top row: w/o Ours; bottom row: w/ Ours. Our method has a significant improvement over PCM in terms of both structure and color balance.

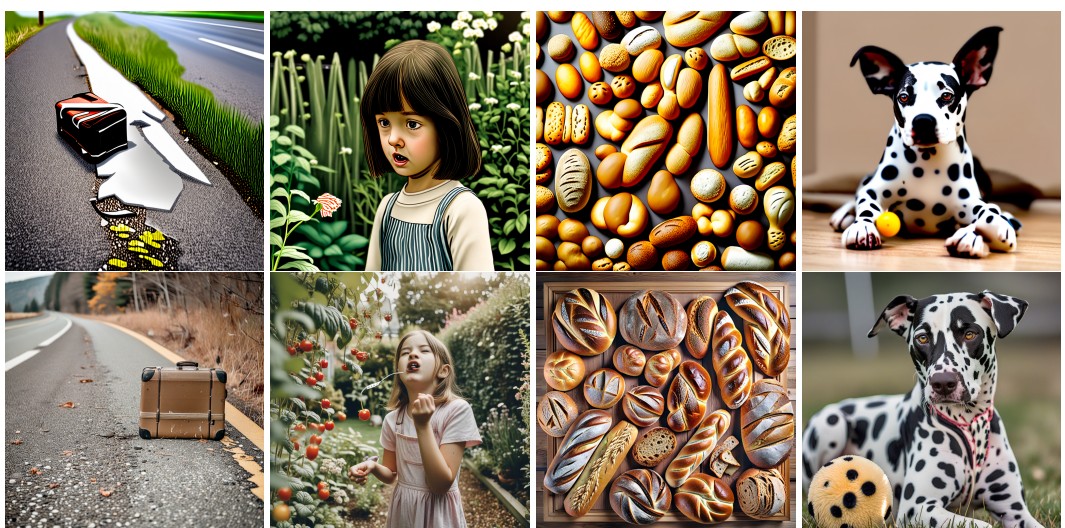

Figure 10: Qualitative comparison of TCD (Zheng et al., 2024) based on Stable Diffusion XL. Top row: w/o Ours; bottom row: w/ Ours. It is obvious that when the CFG value is high (CFG=7.5), TCD has obvious overexposure problem. Our method solves this problem.

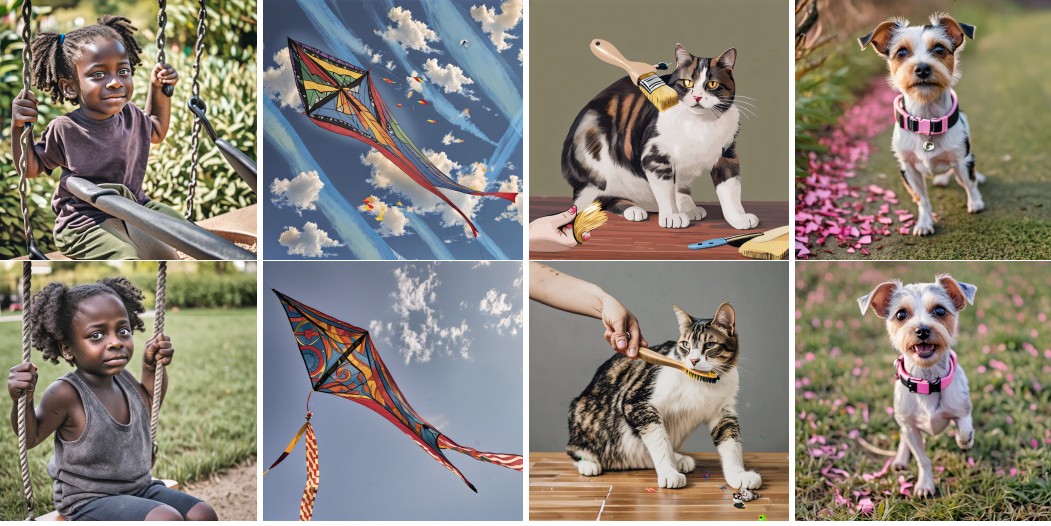

Figure 11: Qualitative comparison of TDD (Wang et al., 2025) based on Stable Diffusion XL. Top row: w/o Ours; bottom row: w/ Ours. Although TDD alleviates the exposure problem, some structures are still confusing. The result structure generated by our method is more reasonable and consistent.

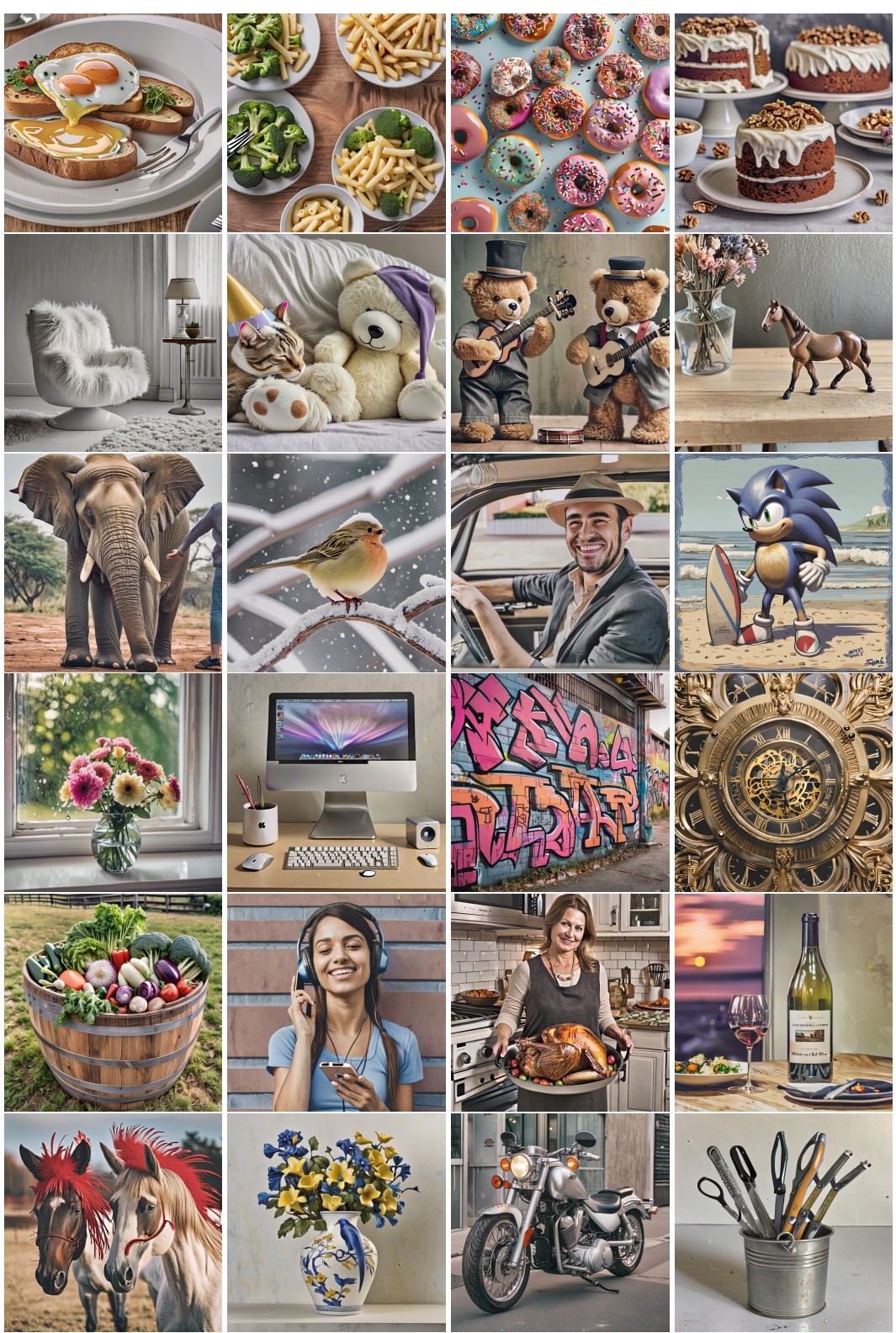

Figure 12: More generation results of ours based on Stable Diffusion XL.