# OpenReview forum: "Adaptive Sampling Scheduler"
_ICLR.cc/2026/Conference — Submitted to ICLR 2026_

### Official Review · Reviewer_tw27 · 2025-10-16

**Soundness:** 2
**Presentation:** 3
**Contribution:** 2
**Rating:** 4
**Confidence:** 4

**Summary:**

This paper proposes an adaptive sampling scheduler for consistency distillation frameworks, with its core design focused on dynamic timestep selection. The authors define the rate of signal change through Equation 7, using this metric to assess the importance of timesteps for selection during distillation. Additionally, a smoothing mechanism for the CFG method is introduced to further improve the quality of the generated output. The authors validate their approach across a variety of scenarios.

**Strengths:**

+ The design approach, which incorporates the properties of forward diffusion, is highly meaningful as it considers the impact of the dynamic properties of the diffusion process.

+ Validation across various consistency model frameworks demonstrates the method's ability to generalize effectively.

**Weaknesses:**

My primary concerns with this paper are as follows:

+ **The meaning of Equation 7 is not entirely clear.** The authors define it as "the rate of signal change," but the numerator appears to be the derivative of the log SNR with respect to timestep t, while the denominator represents the maximum value. However, the specific physical significance of this definition is not thoroughly discussed—why is this definition accurate? The authors provide only a basic analysis in Figure 2. Therefore, I suggest that the authors elaborate further on the significance of Equation 7 and provide a more compelling analysis to justify its use.

+ The conclusion that different timesteps contribute differently to the diffusion model has been extensively explored in prior works. For example, it has been observed that timesteps with high SNR are more challenging to train [1,2,3], and P2 [4] emphasizes the greater importance of intermediate timesteps—findings that are closely aligned with the motivation of this paper. I recommend that the authors review these important relevant works to highlight the novel contributions of this paper.

[1] Denoising Task Difficulty-based Curriculum for Training Diffusion Models. ICLR-2025

[2] Beta-tuned timestep diffusion model. ECCV-2024

[3] A closer look at time steps is worthy of triple speed-up for diffusion model training. CVPR-2025

[4] Perception Prioritized Training of Diffusion Models. CVPR-2022

**Questions:**

1. All experiments are built upon and evaluated with the old SD backbone, a model that is outdated and no longer representative of the current state of the field. Given the rapid shift toward transformer-based architectures like DiT, MMDiT, FLUX, and SD3, it is essential to validate the proposed method on these more advanced models. Without this, the experimental claims are difficult to generalize or take seriously.

2. The latest work Meanflow [1] has already achieved high-quality generation in a single step. Could the authors discuss the respective advantages of each approach?

[1] Mean Flows for One-step Generative Modeling. NeurIPS-2025

---

### Official Review · Reviewer_u8ae · 2025-10-20

**Soundness:** 2
**Presentation:** 1
**Contribution:** 2
**Rating:** 2
**Confidence:** 4

**Summary:**

This paper proposes an importance-based consistency distillation for diffusion models, which changes the timestep partitioning used in consistency distillation from equal partition to "importance partition" with more timesteps in the region of greater distributional change (in terms of signal-to-noise ratio). The authors also proposes a $\gamma$-$I$ sampler, tailoring the existing $\gamma$-sampler to their setting, and a couple of heuristics to mitigate the CFG exposure issue in text-to-image diffusion models. The proposed methods show consistent improvenment over exsiting Stable Diffusion-based distillation benchmarks.

**Strengths:**

The proposed methods, especially the sampling scheduler in consistency training, show steady improvements on quantitative evaluation on MSCOCO, compared with other existing consistency distillation methods (Tables 1 & 2). The performance boost against existing methods except TDD is also qualitatively obvious (Figure 5). The paper essentially proposes a time-reparameterization of diffusion models so that the SNR changes linearly, which is an interesting perspective.

**Weaknesses:**

While the underlying idea of the proposed method is interesting, the manuscript can be substantially improved in many aspects:
- [W1] **Reproducibility.** It seems very difficult to reproduce the proposed methods from the paper.
    - [W1-1] $T_I$ and $T_E$ in Eq. 8 are not defined mathematically, and I do not think the authors' explanation in natural language (e.g., "timstep with the maximum importance in diferent intervals") is enough.
    - [W1-2] In Figure 1 there are multiple timesteps appearing such as $t_n, t_{n+1}$ and the target steps $t_e, t_e'$. I could not find any definition of them. How can they be derived from Section 5?
    - [W1-3] There are basically no explanation on training in Section 6. What datasets were used for training? What is "PCM + Ours" for example (is it finetuned from PCM or trained with PCM from scratch with proposed timesteps)? They are not explained even in the Supplementary Material.
- [W2] **Presentation.** The paper is not easy to read, and the presentation is not very consistent throughout the paper.
    - [W2-1] **SNR** is referred to as an important factor of the method in Introduction (and Conclusion), but it is never mentioned in Section 2 to Section 6. Similarly, "**deterministic-stochastic target**" seems to be describing the feature of the proposed sampling scheduler, it only appears in Introduction and is never discussed in the method section; I do not see how the description in Section 5 is related to this concept or Figure 1 (please see [W1-2] as well).
    - [W2-2] The method section (Section 5) is hard to read. The four different methods (adaptive sampling scheduler, $\gamma$-$I$ sampling, color balance, clipping) are explained in a single paragraph. Moreover, **"clipping" is even not mentioned in Section 5**. I assume the L294-L300 explains the proposed clipping technique, but it is impossible to reproduce the method from the description.
- [W3] **Contribution.** The framing of the contribution is not very clear.
    - [W3-1] If the proposed method can be combined with any existing consistency distillation techniques as in Figure 5, it is worth discussing. At the current stage of the manuscript, the position of the paper is unclear; is it a method to boost existing distillation methods, or is it a stand-alone distillation method etc. The title of the manuscript also fails to describe the contribution.
    - [W3-2] While the performance improvement from the methods except TDD is impressive, the improvement from TDD is marginal. The authors should discuss the algorithmic/performance difference between the proposed methods and TDD more seriously, since $(\text{TDD})-(\text{Other methods})\approx (\text{Proposed method}) - (\text{Other methods})$ in performance. In addition, TDD is framed as a method taking balance between efficiency and adaptability around L126, and I think this framing also applies to the proposed method; I wonder how the methods in Section 2 and the proposed method (at least the methods compared in the experiments) are different in terms of training efficiency. The current report does not distinguish the training inefficiency from the incapability of learning few-step sampling.
    - [W3-3] I do not see the relevance of clipping and color balance in this paper. It seems like combining two or three mini-papers on diffusion models, which makes the contribution of the paper ambiguous.

**Questions:**

- [Q1] Related to [W1], what are the mathematical definitions of $T_E$ and $T_I$ and what does the actual algorithm look like given those sets (pseudocode is also fine) ?
- [Q2] Related to [W3-2], I'm not sure whether the performance boost is only valid with limited budget or the proposed method is inherently better than existing methods when the loss converges. Do the authors have any idea on that?
- [Q3] (Just a comment) Why computing importance and divide the timesteps into just two sets? I think placing the timesteps so that the SNR (or log-SNR) changes in the constant speed is more reasonable.

---

### Official Review · Reviewer_Xe4N · 2025-10-26

**Soundness:** 1
**Presentation:** 1
**Contribution:** 2
**Rating:** 2
**Confidence:** 3

**Summary:**

The paper introduces an approach to enhance consistency distillation methods in diffusion models for image generation. It proposes an adaptive sampling scheduler that dynamically selects timesteps based on their computed "importance". Experimental results demonstrate significant performance improvements, particularly with fewer sampling steps (2, 4), outperforming baselines like LCM, PCM, TCD, and TDD.

**Strengths:**

- A method that can be additionally applied with ease to a number of consistency distillation methods, and shows significant performance improvement over baselines. Qualitatively also show significant improvements.

**Weaknesses:**

- This paper is not well written, and very hard to follow; mainly due to notations that are used without defining, notations significantly abused, and lots of design choices that are not justified.
- To list some:
  - Figure 2 shows $T=0$ to $T=999$ in DDPM. $T$ was defined to be a total number of diffusion timesteps, what does this figure mean? Does it mean $t$ instead of $T$?
  - How does Eq (7) represent SNR? What is the motivation of the construction of this "Importance"? How do we know that how figure 2 changes correspond to this importance? This seems very arbitrary and not well explained.
  - $T_I$ is defined to be: timestep with the maximum importance in different intervals. While $T_I$ seems to be a set, what is this "different intervals"?
  - What is original equidistant sampling timestep $T_E$? Is this a set $\{1,2,...,T\}$?
  - $\theta$ is both used for $f_\theta$ and a hyperparameter for adaptive sampling: which is confusing. Why setting $\theta=0.7$ is a reasonable choice?
  - What is $T_n$?
  - How is Figure 4(a) related to Eq (8)? Where is $T_I$ in Figure 4(a)? It is not clear from the figure whether $T_E$ and $T_{as}$  are sets.
  - If $\gamma$ sampler is an algorithm central in the development of the proposed $\gamma-I$ sampler, it should have been explained in Preliminaries.
  - How is $\gamma-I$ sampler different from $\gamma$ sampler?
  - As far as I understand, $\gamma-I$ sampler is the sampler that uses $\gamma$-sampler when $t_n \in T_E$ and uses $R_t$ instead of $\sqrt{(1-\gamma)^2}$ when $t_n\in T_I$. Is this correct? It is very hard to infer this, as there is no textual description at all on this, and equations somehow have different indices $n+1$ and $n$ for these two cases. The text says we need to look at figure, but figure also does not have much information.
  - What is "varying degrees of exposure issues"?
  - What is the solutions offered by Saharia et al and Lu et al?
  - Why $\alpha,\beta=0.5$ a reasonable choice? and it is very confusing to have $\alpha,\beta$ and $\alpha_t,\beta_t$ at the same time with very different meanings.
  - It is very confusing to have Eq. (11) with equalities, as those are program statements, not equations.

- Overall, the paper shows strong experimental results, but provides so little motivations on the heuristical design of the proposed approach. Experimental results are mostly about the performance metrics, not about why these works. Moreover, the proposed method itself is only summarized in a single page, which is very poorly written that it is almost impossible to fully understand.

**Questions:**

See Weaknesses.

---

### Official Review · Reviewer_hCpJ · 2025-10-31

**Soundness:** 2
**Presentation:** 2
**Contribution:** 2
**Rating:** 2
**Confidence:** 3

**Summary:**

This paper proposes an Adaptive Sampling Scheduler for consistency-distilled diffusion models. The key idea is to compute the **importance of each timestep** based on the rate of change in the Signal-to-Noise Ratio (SNR), and to sample timesteps non-uniformly, allocating denser sampling in regions where the SNR changes more rapidly. The authors argue that existing deterministic or stochastic schedulers ignore such non-uniform importance and hence fail to allocate computation efficiently.

Empirically, the method reportedly improves generation quality across several few-step sampling settings (2–8 steps) on SDXL and SD-v1.5 backbones, especially under high CFG scales.

**Strengths:**

* The paper contributes to the growing literature on **sampling efficiency and timestep scheduling** in consistency distillation, a topic of active research interest.
* The formulation of timestep “importance” through SNR change rate provides an interpretable heuristic criterion that can be implemented easily in practice.
* The empirical gains (especially in the extreme few-step regime) show potential practical relevance for the consistency model.

**Weaknesses:**

I have marked particularly important points with a star (★).

★ **Misleading Terminology — “Adaptive”**
This is a personal opinion, but the term *adaptive* in “Adaptive Sampling Scheduler” is misleading. Despite the name, the scheduler is **not adaptive in the learning or inference sense**, i.e., it does not adjust dynamically to data, training feedback, or model states. The so-called adaptivity arises solely from a **pre-computed, deterministic heuristic** derived from the fixed noise schedule.

Thus, the method is better characterized as a *static, SNR-weighted timestep scheduler* rather than an adaptive one.

**Literature review**
Non-uniform or importance-weighted timestep sampling has been studied extensively. Relevant prior works have proposed to:

* Heuristic: Log-normal distribution with respect to SNR, e.g., Elucidating the Design Space of Diffusion-Based Generative Models (https://arxiv.org/abs/2206.00364), SD3 (https://arxiv.org/abs/2403.03206)
* Theory: Equalize conditional entropy across timesteps (https://arxiv.org/abs/2504.13612v3; https://arxiv.org/abs/2211.15089).

★ **Experimental Scope and Baseline Fairness**
Experiments are limited to **text-to-image** (T2I) diffusion models (SDXL, SD-v1.5). Given the general claim of universality across consistency distillation methods, broader validation is expected. I strongly recommend including **CIFAR-10** and comparing against canonical methods such as **sCM** (https://arxiv.org/abs/2410.11081) and **ECM** (https://arxiv.org/abs/2406.14548) using the proposed scheduler. Please compare the method's performance based on FID, or FD-DINO (https://arxiv.org/abs/2306.04675). This would demonstrate whether the gains persist beyond high-capacity text-conditioned models.

* The paper does not clarify whether baselines were re-tuned or simply reused. The reported numbers for PCM/TCD/TDD appear substantially worse than typically published, raising concerns about fair comparison.

**Questions:**

Ablations on smoothing and color balance and I-$\gamma$-sampler are presented only qualitatively. Please provide quantitative comparisons (e.g., FID) for these components.

---

### Meta-Review · Area_Chair_mX9G · 2026-01-06

**Summary:**

This paper proposes an adaptive, importance-based timestep scheduler for consistency distillation in text-to-image diffusion models. Timestep “importance” is estimated from the rate of change of the signal-to-noise ratio (SNR). The scheduler then allocates denser sampling to regions with larger distributional change, instead of using equal partitions. The method also introduces a tailored sampler and simple heuristics to mitigate CFG exposure issues. It includes a CFG smoothing strategy to improve output quality. Experiments on Stable Diffusion v1.5 and SDXL show consistent quality improvements in few-step generation (2–8 steps). The method often outperforms prior distillation baselines such as LCM, PCM, TCD, and TDD. Gains are especially strong at very low step counts and high CFG scales.

**Reviewer Concerns:**

Reviewers raised the following concerns:
1. Terminology: Although the authors describe the method as “adaptive time scheduling,” the schedule is not dynamically adjusted per input; it is precomputed and fixed.
2. Evaluation scope: Insufficient test cases beyond text-to-image generation (e.g., CIFAR-10), missing comparisons with important baselines (sCM, ECM), and limited evaluation metrics (e.g., FID, FD-DINO).
3. Related work: Prior literature on non-uniform or importance-weighted timestep sampling for diffusion models is not adequately discussed.
4. Clarity: The method description lacks sufficient detail and is difficult to follow, leading to many questions about the approach. It is also unclear whether the proposed method can be applied to existing consistency distillation techniques.

**Reviewer Scores:**

This submission received all negative scores (2, 2, 2, 4). The authors did not submit a rebuttal. Most reviewers raised concerns about the clarity of the writing. They also noted issues with terminology, as well as a lack of literature review and evaluation.

---

### Decision · Program_Chairs · 2026-01-26

Reject